# Tissue Oximeter with Selectable Measurement Depth Using Spatially Resolved Near-Infrared Spectroscopy

**DOI:** 10.3390/s21165573

**Published:** 2021-08-19

**Authors:** Masatsugu Niwayama, Naoki Unno

**Affiliations:** 1Graduate School of Medical Photonics, Shizuoka University, Hamamatsu 432-8561, Japan; 2Division of Vascular Surgery, Hamamatsu University School of Medicine, Hamamatsu 431-3192, Japan; unno@hama-med.ac.jp

**Keywords:** spatially resolved spectroscopy, oximetry, measurement sensitivity, hemoglobin

## Abstract

Tissue oxygenation sensing at a few millimeters deep is useful for surgical and postoperative management. However, the measurement sensitivity at each depth and the proper sensor combination have not been clarified. Here, the measurement characteristics of oximetry by spatially resolved near-infrared spectroscopy were analyzed using Monte Carlo simulation and phantom experiment. From summing the sensitivities of each depth, it was quantitatively found that the measurement sensitivity curve had a peak, and the measurement depth can be adjusted by combining the two distances between the light source and the detector. Furthermore, the gastric tissue was 10–20% smaller in terms of measurement depth than the skin-subcutaneous tissue. A miniaturized oximeter was prototyped so that it could be used in combination with an endoscope or laparoscope. The optical probes consisted of light emitting diodes with wavelengths of 770 nm and 830 nm and photodetectors located 3 to 30 mm from the light source. Phantom experiments using the probes demonstrated the tendency of theoretical analysis. These results suggest the possibility of measuring tissue oxygen saturation with a selectable measurement depth. This selectable method will be useful for obtaining oxygenation information at a depth of 2–5 mm, which is difficult to measure using only laparoscopic surface imaging.

## 1. Introduction

The latest endoscopic and laparoscopic techniques are used for confirming the morphology and tissue properties of living tissues in surgery and diagnosis [1,2]. Studies have also been conducted to assist surgery by adding information on blood flow to the monitoring of laparoscopic surgery in the surface layer of tissues to about 1 mm deep. Spectroscopic tissue oximetry using diffused near-infrared light is a technique that measures hemodynamics from the tissue surface to a depth of ~10 mm. The combination of the laparoscope and the oximeter complements the information about different depths. General oximeters, pulse oximeters, and tissue oximeters are widely used. Pulse oximetry [3], which detects arterial blood pulsation, is ideal for monitoring respiratory status. Tissue oximetry for whole blood is useful for observing tissue oxygenation related to tissue viability. Three major approaches used for measuring tissue oxygen saturation are time-resolved [4], frequency-domain [5], and spatially resolved spectroscopy (SRS) [6,7]. The time-resolved method determines the scattering and absorption coefficients from the time-resolved profile using a femtosecond pulse laser and a streak camera. The frequency-domain method measures the scattering and absorption coefficients using laser light modulated in the gigahertz order. In the spatially resolved method, continuous light is used. The effective absorption coefficient is obtained from the spatial profile of the light intensity, and the absorption coefficient is calculated by assuming the scattering coefficient. Since the oxygen saturation is found by calculating the concentration ratio, it can be obtained without knowing the scattering coefficient. Since the spatially resolved method can be configured with a simple system, it is suitable for miniaturization, cost reduction, and disposability, and the application fields of commercially available tissue oxygen oximeters are gradually expanding. The SRS tissue oximeter with a source–detector distance of 20–40 mm has been applied to the brain and muscles [8] and has recently been used for the indirect assessment of brown adipose tissue [9]. These measurements are generally interpreted as measuring information at a depth of about half the source–detector distance. On the other hand, in recent years SRS devices with source–detector distances reduced to 10 mm or less have been developed to measure surface layers [10,11,12]. The sensor can obtain information at depths of 4–5 mm. The combination of the laparoscope and oximeter can be expected to complement the information on the tissue surface with oxygenation that is several millimeters deep. For example, the tissue oximeter has been applied in surgery for gastric tubes [13] and flaps [14], and sensors with source–detector distances of 6 mm and 8 mm have been used. If the measurement depths with smaller or larger sensor configurations are clarified, it can be expected to obtain information at some depths of the organs of interest while viewing the images with those scopes. In a study on the measurement characteristics of oximetry, the optimum measurement distance in a pulse oximeter was investigated [15,16]. However, pulse oximetry is a one-point absorbance measurement, while SRS oximetry is a method that uses the spatial intensity slope of a multipoint measurement. Therefore, the previous findings cannot be used directly because of the difference in the measurement sensitivity distribution. In this study, we examined the measurement sensitivity in the depth direction with the SRS tissue oximeter and the optimum source–detector distance. Additionally, it is unclear whether a difference in measurement depth between the skin-subcutaneous and gastric tissues exist. To comprehensively analyze the difference in measurement depth, we proposed a measurement depth map by combining two source–detector distances. From these findings, a method for making the measurement depth variable was examined.

## 2. Materials and Methods

### 2.1. Theoretical Analysis

Monte Carlo analysis [11,17,18,19] was performed to clarify the measurement sensitivity at each small volume in biological tissue based on radiative transfer theory [19,20,21,22]. First, by assuming a widely used semi-infinite homogeneous model, we analyzed how the change in the sensitivity distribution of SRS depends on the combination of detectors. We assumed a homogeneous medium with the optical properties of the skin that had an average absorption coefficient among the four types of tissue. Next, layered models were used to analyze the light diffusion within the medium of the skin-subcutaneous and gastric tissues. The optical properties and layer thickness of the models were set using values in the literature [23,24,25,26,27,28,29], as shown in Table 1.

The skin absorption coefficient was set from the average of the three studies [23,24,25]. Since the absorption coefficient of hemoglobin at 770 nm and 830 nm is almost the same as that of a tissue oxygen saturation of 65%, the same absorption coefficient was used at both wavelengths. For the scattering coefficient of the skin, properties under no pressure [26] were used. The absorption coefficient of gastric tissue, which varies depending on the lesion stage, was set at 0.03 mm^−1^, which is the average value of the absorption coefficients in the normal and high lesion stages. The models were divided into 0.2 mm voxels to examine the measurement sensitivity of each site. To determine the measurement sensitivity, firstly we recorded the optical path lengths in the voxels sequentially and then the average optical path length was calculated using the formula:*L* = Σ*L_i_I_i_*/Σ*I_i_*.(1)

Here, *L_i_* is the distance that the *i*-th photon group that reached the detector passed through in the voxel at coordinates *x*, *y*, and *z*, and *I_i_* is the light intensity when the photon group reached the light detector voxel. The relationship between the average optical path length *L* and the change in spatial slope Δ*S* is expressed by the product of the change in the absorption coefficient Δ*μ_a_* and the difference between the two optical path lengths as follows [10]:Δ*S*(*x*, *y*, *z*) = Δ*μ*_*a*_(*x*, *y*, *z*) {*L_B_*(*x*, *y*, *z*) − *L_A_*(*x*, *y*, *z*)}.(2)

Here, subscript *A* corresponds to the light detected near the light source among the two detectors for using the SRS method, while *B* corresponds to the far side. The distances between the light source and the detectors located at *A* and *B* were *ρ*_1_ and *ρ*_2_, respectively. In general continuous-wave NIRS, the change in absorbance is Δ*μ_a_ L*, and *L* is the measurement sensitivity [30]. Similarly, the measurement sensitivity in SRS is defined as {*L_B_*(*x*, *y*, *z*) − *L_A_*(*x*, *y*, *z*)} (mm). The measurement sensitivity at depth *z* is calculated by adding the measurement sensitivities of all voxels at depth *z*. Therefore, the measurement sensitivity of a certain layer is obtained by adding the values of all voxels constituting that layer.

The measurement depth was analyzed using the sensor arrangement of the probe used in this study, and the change in the measurement depth due to the combination of *ρ*_1_ and *ρ*_2_ was exhaustively investigated as follows. First, *ρ*_2_ was fixed and *ρ*_1_ was changed from 0.5 mm to *ρ*_2_ − 0.5 mm, then curves of the relationship between depth and measurement sensitivity were obtained. Then, the curves were normalized to clarify the position of the maximum value. The same calculations were performed when *ρ*_2_ is changed to 5, 10, 20, and 30 mm, and we created a map of how the peak of the measurement depth changed depending on *ρ*_1_ and *ρ*_2_. In the analysis with these many combinations, the voxel size was set to 0.5 mm to reduce the amount of physical random access memory used.

### 2.2. Spatially Resolved Oximeter

Since the SRS method calculates the optical properties from the spatial intensity slopes of multiple wavelengths, the SRS oximeter requires multiple wavelengths and multiple source–detector pairs. To enable the combined use of oximeters with a laparoscope, we designed a small optical probe of width 10 mm that can be inserted through the laparoscopic port. A rubber and resin frame was attached for stable measurement. Four optical probes with two detectors were developed to measure different depths, as shown in Figure 1. 

The photodetectors of the optical probes were located 3–30 mm from the light source consisting of light emitting diodes (LEDs) with wavelengths of 770 nm and 830 nm. SRS measurement did not require high-frequency elements used for time-domain or frequency-domain measurements, and it was only necessary to be able to perform the continuous light measurement of each wavelength intermittently every 100 ms. Therefore, several tens of kHz were sufficient for the bandwidth values of the light source, driver, photodetectors, and amplifiers. A system consisting of these general-purpose circuit elements was realized at low cost and in a small size. Low-cost probes can serve as disposables and be used to eliminate the risk of infection. The photodetectors shown in Figure 1a–d are a light-to-digital sensor (Texas Instruments Inc., Dallas, TX, USA, OPT3002) and a photodiode (Hamamatsu Photonics Inc., Hamamatsu, Japan, S2387-16R), respectively. The signal of the light-to-digital sensor was taken into the microcomputer using the inter-integrated circuit (I^2^C) communication protocol. The photocurrent of the photodiode was converted from current to voltage by an operational amplifier and inputted into a microcomputer. The oximeter was powered by two AA batteries, and the LED light intensity was stable for 12 h using a low noise voltage linear regulator and a 30 mA constant current regulator. It was also confirmed that there was no temperature drift of light intensity. Each optical probe had two detectors and *ρ* is the average value of the two source–detector distances *ρ*_1_ and *ρ*_2_ (*ρ*_1_ < *ρ*_2_). The spatial slope *S* was calculated as ln (*I_A_*/*I_B_*)/*ρ*, where *I_A_* and *I_B_* were the intensities at the two positions and *μ*_s_′ was the reduced scattering coefficient. In the homogeneous structure model, the absorption coefficient was calculated using Equation (3), which was derived from the diffusion approximation [8].
(3)$$\mu_{a}=\frac{1}{3\mu_{s}^{\prime }}{\left(S-\frac{2}{\rho }\right)}^{2}.$$

In this study, the short source–detector distances of some probes can lead to errors due to the diffusion approximation of Equation (3) and the inhomogeneity of the model. Therefore, a reference table of the relationship between the absorption coefficient and the spatial slope was created using the Monte Carlo simulation, based on the radiative transfer theory to minimize the errors.

The oxygenated hemoglobin concentration (O_2_Hb) and the deoxygenated hemoglobin concentration (HHb) were calculated using the absorption coefficients *μ*_a770_ and *μ*_a830_ of the two wavelengths.
(4)$$\left[O_{2}\mathrm{Hb}\right]=\frac{1}{k}\left(\varepsilon_{830}^{\mathrm{HHb}}\mu_{a770}-\varepsilon_{770}^{\mathrm{HHb}}\mu_{a830}\right),$$
(5)$$\left[\mathrm{HHb}\right]=-\frac{1}{k}\left(\varepsilon_{830}^{O_{2}\mathrm{Hb}}\mu_{a770}-\varepsilon_{770}^{O_{2}\mathrm{Hb}}\mu_{a830}\right),$$
(6)$$k=\varepsilon_{770}^{O_{2}\mathrm{Hb}}\varepsilon_{830}^{\mathrm{HHb}}-\varepsilon_{830}^{O_{2}\mathrm{Hb}}\varepsilon_{770}^{\mathrm{HHb}},$$
where $\varepsilon_{770,830}^{O_{2}\mathrm{Hb}}$ and $\varepsilon_{770,830}^{\mathrm{HHb}}$ are the extinction coefficients of O_2_Hb and HHb, respectively, at wavelengths of 770 and 830 nm [31]. Blood volume totalHb is the sum of O_2_Hb and HHb, and regional oxygen saturation rSO_2_ was calculated using the formula:(7)$${\mathrm{rSO}}_{2}=\frac{\left[O_{2}\mathrm{Hb}\right]}{\left[O_{2}\mathrm{Hb}\right]+\left[\mathrm{HHb}\right]}.$$

Phantom experiments were performed using the oximeter connected to the above probe. To experimentally investigate the sensitivity in the depth direction, the phantom with a thin blood layer in a homogeneous fat-like tissue was used. The blood-containing layer was made by solidifying 1.2% Intralipos (Otsuka Pharmaceutical Co., Ltd., Tokyo, Japan) [32] and 1% bovine blood (Cosmo Bio Co., Ltd., Tokyo, Japan) with agar. The blood was fully oxygenated with oxygen gas and added after boiled agar had cooled below 104 degrees Fahrenheit. A fat-like scattering medium layer (1.2% Intralipos with agar) was placed above and below the blood layer, and the thickness of the scattering layer and the type of probe were changed to compare with the theoretical tendency.

## 3. Results

Figure 2 shows the measurement sensitivity distributions for four source–detector pairs. The notation “3–5 mm pair” means that the spatial slope was calculated using the intensities at the source–detector distances of 3 and 5 mm, respectively. The red-yellow-green color represents positive sensitivity, while the cyan-blue color represents negative sensitivity. Suppose the absorption coefficient of tissue with negative sensitivity increases, the spatial slope and the calculated absorption coefficient decrease. In Figure 3, the sensitivities of the same depth are summed to evaluate the sensitivity of each depth. In Figure 2, the sensitivity of the 0.2 mm surface layer had both large positive and negative values, but most of the positive sensitivities were canceled by negative values, and the sum of the surface layer sensitivities in Figure 3 was low. The peak depth of measurement sensitivity with a pair of 3–5 mm was 2 mm, and the peak became deeper with an increase in the source–detector distance. Table 2 summarizes the peak depth of measurement sensitivity, the depth range at half maximum, and the depth reduced to 10% of the maximum. This sensitivity information about the depth will be useful for selecting an optical probe for measuring any tissue depth of interest.

Figure 4 shows the map of the effect of the measurement depth and the combination of *ρ*_1_ and *ρ*_2_ on the normalized measurement sensitivity in the skin-subcutaneous tissue model. The measurement sensitivity was represented using blue for negative, yellow for positive, and red for the peak value (1.0). The results show that the depth *D_p_*, which indicates the peak of the measurement sensitivity in SRS, increased with an increase in *ρ*_1_ and *ρ*_2_. For example, when *ρ*_1_ was about half of *ρ*_2_, *D_p_* was approximated to be about one fifth of the length of *ρ*_2_. Since both *ρ*_1_ and *ρ*_2_ affected the values of *D_p_*, we expressed *D_p_* (mm) as a function of *ρ*_1_ (mm) and *ρ*_2_ (mm) by regression analysis.
(8)$$D_{p}=0.090\;\rho_{1}+0.079\;\rho_{2}+0.85.$$

The map of the normalized measurement sensitivity in the gastrointestinal tissue model is shown in Figure 5. The measurement depth is slightly shorter than that of the skin-subcutaneous tissue model, and Equation (9) was obtained as a regression equation for *D_p_* (mm), *ρ*_1_ (mm), and *ρ*_2_ (mm).
(9)$$D_{p}=0.041\;\rho_{1}+0.073\;\rho_{2}+0.76.$$

These results show that the differences in *D_p_* between the skin-subcutaneous and gastric tissues were 10% to 20% when the distance *ρ*_2_ was 10 mm or less and that the oximeters for the two tissues produces similar measurement sensitivity distributions. The measurement depth can be changed in the range of 1–5 mm by selecting *ρ*_1_ and *ρ*_2_ according to the measurement depth of interest and the tissue type.

Figure 6 shows the measurement results when the depth of the blood layer changed in the phantom experiment. As shown by the dashed circles in Figure 6, the peak value changed due to the sensor separation of the probe. Since the blood layer was oxygenated to almost 100%, it follows that the oxygen saturation is high at a depth where the measurement sensitivity is high. Furthermore, the peak depth of this phantom experiment was consistent with the peak obtained in the theoretical measurement sensitivity.

## 4. Discussion

As shown in Figure 2, the positive and negative values of the surface layer sensitivity were large, but the values added in the same depth layer were canceled, as shown in Figure 3, and the surface layer sensitivity was reduced. Therefore, if the tissue is homogeneous in the direction of the source–detector axis, the result obtained is the same as that in theory. However, for inhomogeneity, such as a thick blood vessel near only one detector, the spatial slope changes, and an error in the rSO_2_ occurs. For the homogeneity of the surface layer, it would be effective to visually confirm that the conditions near the two detectors are the same, and to measure several times by shifting the probe contact position by 2 to 3 mm. Although analyses with inhomogenously distributed absorbers have been reported [33,34], we assumed that hemoglobin was homogeneously distributed within each layer. Therefore, the results using our model may have less variation in the probability of photon absorption depending on the detection position compared with inhomogeneous modeling. The influence of chromophore inhomogeneity on quantification in various tissues should be further investigated.

In the case of a semi-infinite medium with a long source–detector distance (~30 mm), many photons return to the detector after reaching a depth of ~10 mm. Therefore, in measuring the 7 mm gastrointestinal tissue, the light intensity was attenuated, and the sensitivity of the deep part was reduced due to the emission of the light source from the opposite surface. To measure thin tissues, a 3–5 mm pair or a 6–8 mm pair are suitably used, since they have a high measurement sensitivity in the surface layer.

Next, we describe the absolute value of rSO_2_ in Figure 6. When the scattering layer not containing blood on the surface was thick and the source-detector separation was short, the information on the blood layer was extremely small but the rSO_2_ value was about 70–80%. Since the spatial slope in the SRS method was approximately proportional to $\sqrt{3\mu_{s}^{\prime }\mu_{a}\;}$ [6] when the absorption coefficient was extremely small—such as water or fat—it implies that the output value of the oximeter was more influenced by the scattering coefficient spectrum than the blood absorption spectrum. The data at a depth of 1.5 mm in Figure 6 correspond to, for example, the dermis of plantar tissue, and the data at 2.5 mm correspond to the subcutaneous tissue. The results also experimentally suggest that the appropriate sensor pair depends on the depth of interest.

By understanding the above-mentioned error factors (surface tissue inhomogeneity, reduced blood volume, and thin tissue) and applying the sensitivity curves obtained in this study, it will be possible to measure the tissue oxygen saturation with selectable measurement depth. This method will be useful in obtaining oxygenation information at a depth of 2–5 mm, which is difficult to measure when using only an endoscope or laparoscope. The skin-subcutaneous tissue measurement [12] can be applied to monitor the oxygenation of the planer tissue and confirm the therapeutic effect in endovascular therapy for patients with chronic limb-threatening ischemia. The evaluation of blood perfusion can be more reliable by integrating information at several depths. The gastric tube tissue measurement [13] can be used in surgery for esophageal cancer when selecting the gastric anastomosis. The observation of oxygenation at a depth of 2 mm to 5 mm would lead to a reduction in suture failure due to an accurate assessment of tissue viability. The relatively small difference in measurement depth between the skin-subcutaneous and gastric tissues means that the same sensor probe can be used with a slight depth correction in both applications. This leads to improved convenience and cost reduction by simplifying the device in applications in various tissues.

## 5. Conclusions

In this study, the measurement depth of oximetry by spatially resolved near-infrared spectroscopy was examined using Monte Carlo analysis and a phantom experiment. From the results of the sensitivities for each depth, it was quantitatively found that the measurement sensitivity curve had a peak and that the measurement depth could be adjusted by combining the two distances between the light source and the detector. These results suggest the possibility of measuring tissue oxygen saturation at depths of 2–5 mm with selectable measurement depths. The evaluation of tissue viability could be made more reliable by observing oxygenation at some depths. Furthermore, the gastric tissue was 10–20% smaller in terms of measurement depth than the skin-subcutaneous tissue. This means that the same optical probe can be widely applied by correcting the measurement depth according to the target tissue.

## Figures and Tables

**Figure 1 sensors-21-05573-f001:**
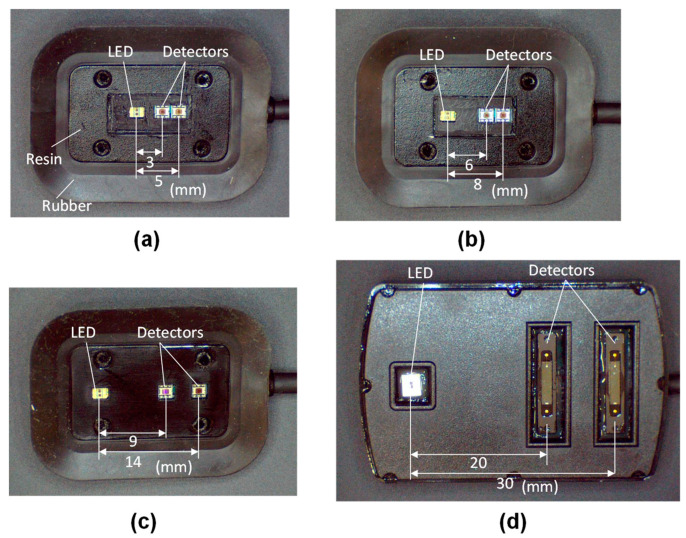
Optical probes with detectors located (**a**) 3 and 5 mm, (**b**) 6 and 8 mm, (**c**) 9 and 14 mm, and (**d**) 20 and 30 mm from the light source (LED).

**Figure 2 sensors-21-05573-f002:**
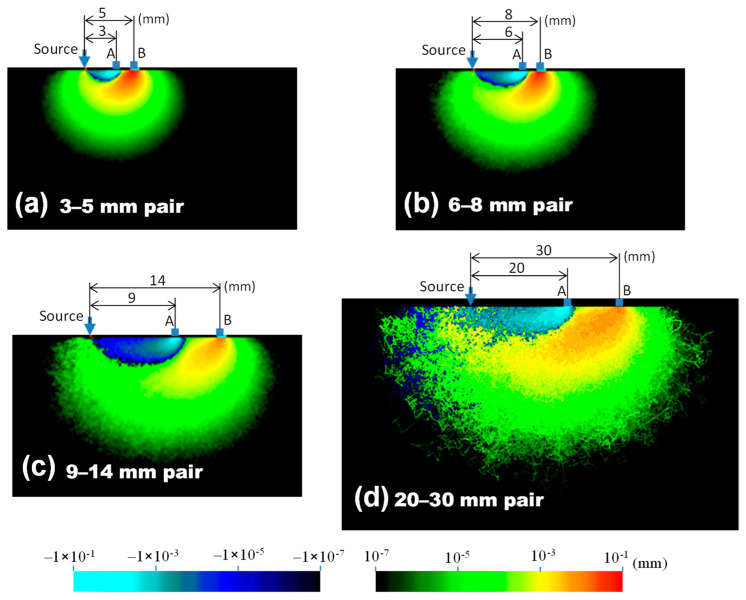
Simulation results for the measurement sensitivity distribution in a homogeneous medium with four SRS probes. The combination pairs for the two detectors of the probe were a (**a**) 3–5 mm pair, (**b**) 6–8 mm pair, (**c**) 9–14 mm pair, and (**d**) 20–30 mm pair.

**Figure 3 sensors-21-05573-f003:**
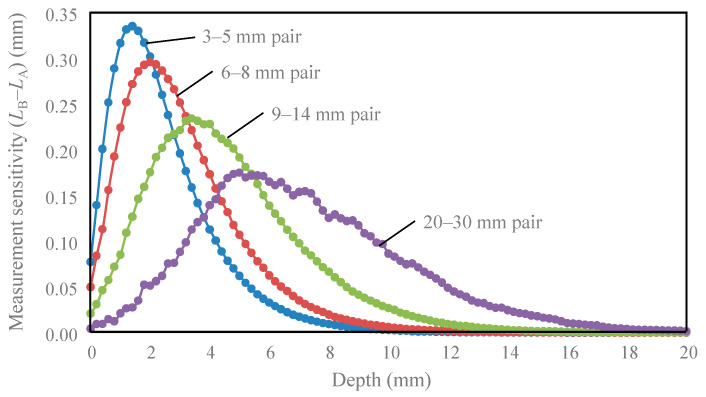
The relationship between measurement sensitivity and measurement depth in the four pairs of two detectors.

**Figure 4 sensors-21-05573-f004:**
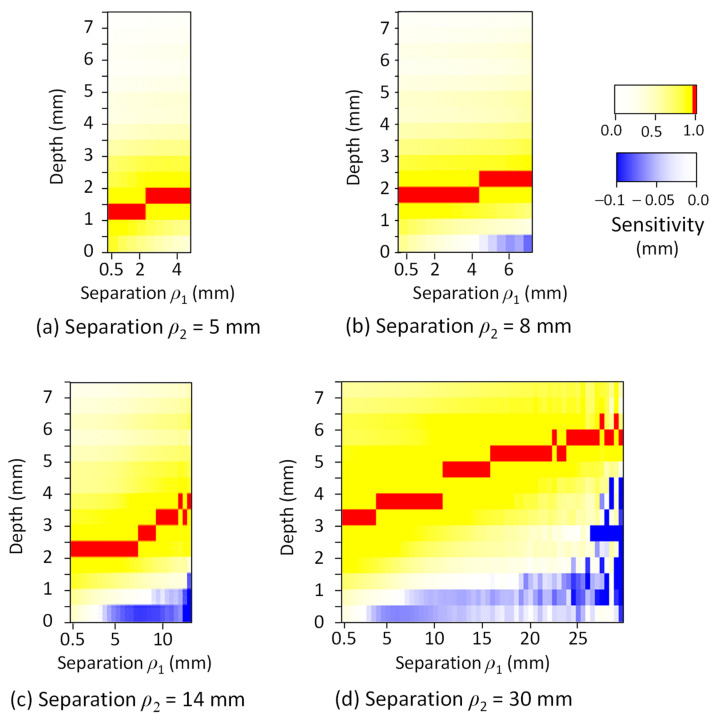
The mapping result of the normalized sensitivity–depth curve in the skin-subcutaneous tissue model. The source–detector separation (far position) *ρ*_2_ was changed to (**a**) 5 mm, (**b**) 8 mm, (**c**) 14 mm, and (**d**) 30 mm.

**Figure 5 sensors-21-05573-f005:**
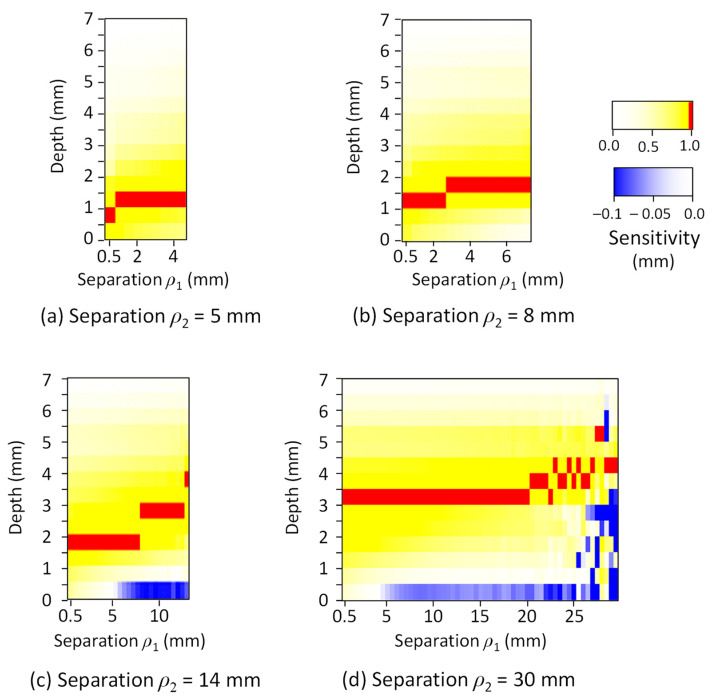
The mapping result of the normalized sensitivity–depth curve in the gastric tissue model. *ρ*_2_ was changed to (**a**) 5 mm, (**b**) 8 mm, (**c**) 14 mm, and (**d**) 30 mm.

**Figure 6 sensors-21-05573-f006:**
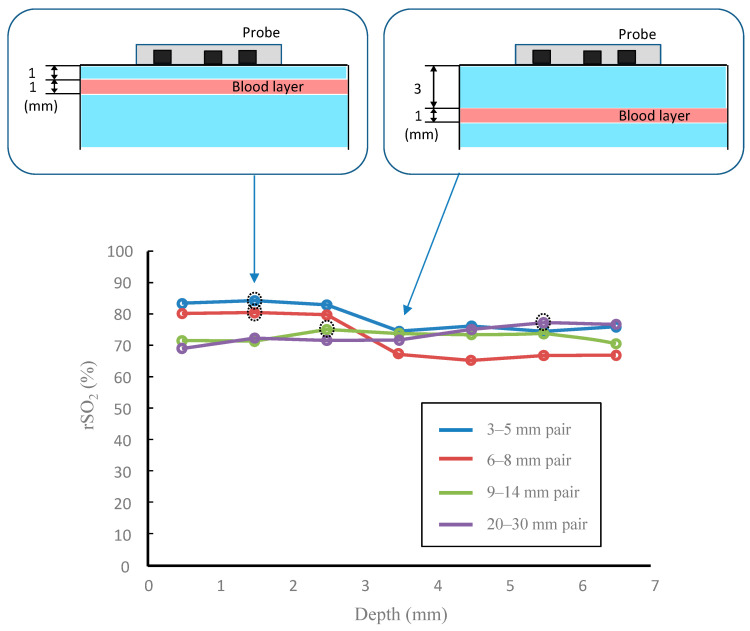
The results of the phantom experiments. Changes in regional oxygen saturation (rSO_2_) with four probes when blood layer depth changes. The dashed circles represent the peak values of each curve. For example, when the blood layer was at a depth of 1–2 mm, the data were plotted at a depth of 1.5 mm.

**Table 1 sensors-21-05573-t001:** Optical properties and layer thickness in simulation models.

Tissue Type	Thickness (mm)	Scattering Coefficient (mm^−1^) 770 nm and 830 nm		Absorption Coefficient (mm^−1^) (770 and 830 nm)	Anisotropic Factor (770 and 830 nm)
Skin	1.5	26	23	0.020	0.95
Fat	2.5	24	22	0.003	0.95
Muscle	20.0	14	13	0.025	0.95
Gastric tissue	7.0	15	14	0.030	0.92

**Table 2 sensors-21-05573-t002:** The depth where the sensitivity is maximum, the range of the depth where the sensitivity halves the maximum, and the depth where the sensitivity is 10% of the maximum obtained in the curves of the measurement sensitivity and measurement depth of the four pairs.

Detector Pair	Peak Depth (mm)	Depth Range of 50% of Peak (mm)	Depth Reduced to 10% of Peak (mm)
3–5 mm	1.4	0.3–3.3	6.0
6–8 mm	2.0	0.5–4.5	7.3
9–14 mm	3.4	1.3–6.5	10.3
20–30 mm	5.0	3.0–9.8	15.0

## Data Availability

Not applicable.

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
