# Peer review of "Tissue Oximeter with Selectable Measurement Depth Using Spatially Resolved Near-Infrared Spectroscopy"

_sensors, 2021, doi:10.3390/s21165573_

Round 1
Reviewer 1 Report
Authors proposed a measurement depth map by combining two source-detector in models of skin-subcutaneous and gastric tissues. Recent studies highlight the persistence of interest in understanding the value of tissue oximetry in general patient care.
The methodology, tables, and figures are clear for readers.
There are not clearly summarized conclusions and clinical usefulness of this results. How optimum source-detector distance or a difference in measurement depth between the skin-subcutaneous and gastric tissues may improve clinical treatment and the usefulness of methods? How does the measurement of depth variable change the laparoscopy methods and further clinical outcomes of patients?
Author Response
(Please see the attachment)
RESPONSE TO REVIEWER 1:
We wish to express our appreciation to the reviewers for insightful comments on our paper. The comments have helped us significantly improve the paper. Green texts are the reviewer's comment, and blue letters are the sentences added to the revised manuscript.
Comment 1: The subject of the manuscript submitted is of special importance for medical applications. Tissue oximetry has been around for many decades, but also novel developments especially also for pulse oximetry must be mentioned, which tackles in particular the oxygenation of arterial blood.
For this also modelling of the radiative transfer in scattering media has been presented, for which also optimal source-detector spacings were determined.
(see Mehrabi M, Setayeshi S, Ardehali SH, Arabalibeik H (2017) Modeling of diffuse reflectance of light in heterogeneous biological tissue to analysis of the effects of multiple scattering on reflectance pulse oximetry. J Biomed Optics 22 (1): 015004;
- Mehrabi, S. Setayeshi, M. G. Maragheh, S.H. Ardehali, H. Arabalibeik, Design of a new reflectance pulse oximeter by obtaining the optimal source-detector space, Optik 168 (2018) 34-45).
Response: Thank you for providing these insights. We have added a description and literature about pulse oximetry and its analysis.
<Revised manuscript>
(Line 32) As general oximeters, pulse oximeters and tissue oximeters are widely used. Pulse oxime-try[3], which detects arterial blood pulsation, is ideal for monitoring respiratory status, and tissue oximetry for whole blood is useful for observing tissue oxygenation related to tissue viability.
(Line 60) As a study on the measurement characteristics of oximetry, the optimum measurement distance in a pulse oximeter was investigated [15–16]. However, pulse oximetry is a one-point absorbance measurement, while SRS oximetry is a method that uses the spatial intensity slope of multipoint measurement. Therefore, the previous findings cannot be used directly because of the difference in measurement sensitivity distribution. In this study, we examined the measurement sensitivity in the depth direction with the SRS tis-sue oximeter and the optimum source-detector distance.
[15] Mehrabi, M. et al., Modeling of diffuse reflectance of light in heterogeneous biological tissue to analysis of the effects of multiple scattering on reflectance pulse oximetry. J. Biomed. Opt. 2017, 22, 015004, doi:10.1117/1.JBO.22.1.015004.
[16] Mehrabi, M. et al., Design of a new reflectance pulse oximeter by obtaining the optimal source-detector space. Optik (Stuttg). 2018, 168, 34–45, doi:10.1016/J.IJLEO.2018.04.039
Comment 2: The abstract also should contain all relevant information, for example the wavelengths of the LEDs used for tissue oximetry.
Response: Regarding the missing information in Abstract, we have added the following sentence:
<Revised manuscript>
(Line 16) The optical probes consist of light emitting diodes with wavelengths of 770 nm and 830 nm and photodetectors located 3 to 30 mm from the light source.
Comment 3: As endoscopic applications had been envisaged, much more focus must be given the smaller source-detector configurations, which is not so limiting for skin-subcutaneous tissue applications. This must be more discussed within the introduction.
Response: We aim to observe the oxygenation of organs at a depth of several millimeters while checking endoscopic or laparoscopic images. We have added a recent application literature and added the need for the smaller and larger source-detector configurations to the introduction.
<Revised manuscript>
(Line 55) For example, tissue oximeter has been applied in surgery for gastric tubes [13] and flaps [14], and sensors with source-detector distances of 6 mm and 8 mm were used. If the measurement depths with smaller or larger sensor configurations are clarified, it can be expected to obtain information at some depths of the organs of interest while viewing the images with those scopes.
[13] Fujita, T. Method for evaluating gastrointestinal blood flow and reducing suture failure in esophageal cancer: Blood flow evaluation using regional SO2 (%) and Total Hemoglobin index (Japanese). In Proceedings of the Proc. of the 81st Annual Congress of Japan Surgical Association; 2019; pp. VW02-2.
[14] Tsuge, I.; Enoshiri, T.; Saito, S.; Suzuki, S. A Quick Evaluation of TRAM Flap Viability using Fingerstall-Type Tissue Oximetry. Plast. Reconstr. Surg. Glob. Open 2017, 5, doi:10.1097/GOX.0000000000001494.
Comment 4: Another important point is that optical parameters of tissue are wavelength dependent, which has not been given with Table 1. In addition, literature must be updated in view of more recent publications with regard to optical constants of skin and layered sections; see the following papers:
Response: As you pointed out, there was a lack of wavelength-specific information and recent papers, so we added it to the text, the table and the references. Since the absorption coefficient of hemoglobin at 770 nm and 830 nm is almost the same at 65% StO2, the same absorption coefficient was used at the two wavelengths.
<Revised manuscript>
(Line 85) The skin absorption coefficient was set from the average of the three literatures. The absorption coefficient of hemoglobin at 770 nm and 830 nm is almost the same at 65% StO2, so the same absorption coefficient was used at the two wavelengths. The value of the scattering coefficient of the skin under no pressure was used.
[24] Bashkatov, A.N.; Genina, E.A.; Kochubey, V.I.; Tuchin, V. V Optical properties of human skin, subcutaneous and mucous tissues in the wavelength range from 400 to 2000 nm. J. Phys. D-Applied Phys. 2005, 38, 2543–2555, doi:10.1088/0022-3727/38/15/004.
[25] Simpson, C.R.; Kohl, M.; Essenpreis, M.; Cope, M. Near-infrared optical properties of ex vivo human skin and subcutaneous tissues measured using the Monte Carlo inversion technique. Phys Med Biol 1998, 43, 2465–2478.
Comment 5: Modeling of radiative transfer with aspects of penetration depths has also been presented many times, and special attention should also be given to
Nasouri B, Murphy TE, Berberoglu H. 2014. J Biomed Opt 19:075003.
Zhang H, Salo D, Kim DM, Komarov S, Tai Y-C, Berezin MY. 2016. J Biomed Opt 21:126006
L.F.A. Douven, G.W. Lucassen, Proceedings of SPIE Vol. 3914, 312 -- 323 (2000)
Response: Regarding modeling of radiation transmission, we have added the three literatures in your comments.
<Revised manuscript>
[20] Nasouri, B.; Murphy, T.E.; Berberoglu, H. Simulation of laser propagation through a three-layer human skin model in the spectral range from 1000 to 1900 nm. J. Biomed. Opt. 2014, 19, 075003, doi:10.1117/1.JBO.19.7.075003.
[21] Zhang, H.; Salo, D.; Kim, D.M.; Komarov, S.; Tai Y.-C.; Berezin, M.Y. Penetration depth of photons in biological tissues from hyperspectral imaging in shortwave infrared in transmission and reflection geometries. J. Biomed. Opt. 2016, 21, 126006, doi:10.1117/1.JBO.21.12.126006.
[22] Douven, L.F.A.; Lucassen, G.W. Retrieval of optical properties of skin from measurement and modeling the diffuse reflectance. SPIE 2000, 3914, 312–323, doi:10.1117/12.388058.
Comment 6: I am missing also the simulation with inhomogenously distributed chromophores as presented in the past (see W. Verkruysse et al., Modelling light distributions of homogeneous versus discrete absorbers in light irradiated turbid media, Phys. Med. Biol. 42 (1997) 51?65; F. Niedorf, H. Jungmann, M. Kietzmann, Noninvasive reflection spectra provide quantitative information about the spatial distribution of skin chromophores, Med. Phys. 32 (5), 1297- 1307 (2005)); due to the vessel structures, the homogenously distributed hemoglobin variants are an approximation which needs testing and which has also been used for the phantom experiments as severe simplification.
Response: The following text and references has been added to the Discussion on inhomogeneity.
<Revised manuscript>
(Line 245) Although analysis with inhomogenously distributed absorbers have been reported [33–34], we assumed that hemoglobin was homogeneously distributed within each layer. Therefore, the results using our model may have less variation in the probability of photon absorption depending on the detection position compared to inhomogeneous modeling. The influence of chromophore inhomogeneity on quantification in various tissues should be further investigated.
Comment 7: The authors used a model with a fat-like scattering medium above and below the blood layer, which is not a realistic scenario for skin simulation (see also modeling as presented by Douven and Lucassen above).
Response: We have fixed the following two points.
(1) Inaccurate expression; “The homogeneous medium assumed the optical properties of the skin.”
(2) Purpose of using fat-like phantom
<Original manuscript >
The homogeneous medium assumed the optical properties of the skin.
<Revised manuscript>
(Line 77) We assumed a homogeneous medium with the optical properties of the skin, which had an average absorption coefficient among the four types of tissue.
(Line 163) To experimentally investigate the sensitivity in the depth direction, the phantom with a thin blood layer in a homogeneous fat-like tissue was used.
Comment 8: I am also questioning the two scenarios as given in Fig. 6, which needs discussion and anatomy support.
Response: We added plantar tissue as an anatomical example in the discussion in Fig.6.
<Revised manuscript>
(Line 257) Next, we describe the absolute value of rSO2 in Fig.6. When the scattering layer not containing blood on the surface was thick and the source-detector separation was short, the information on the blood layer was extremely small, but the rSO2 value at that time was about 70%–80%. Since the spatial slope in the SRS method was approximately proportional to [6] when the absorption coefficient was extremely small, such as water or fat, it implies that the output value of the oximeter was more influenced by the scattering coefficient spectrum than the blood absorption spectrum. The data at a depth of 1.5 mm in Fig. 6 corresponds to, for example, the dermis of plantar tissue, and the data at 2.5 mm corresponds to the subcutaneous tissue. The results also experimentally suggest that the appropriate sensor pair depends on the depth of interest.
Comment 9: I think that also the clarity of presentation must be significantly improved and definitions of the terms used be given when used first. For example, on page 3 parameters ρ1 and ρ2 have been presented but not explained, and on page 4 ρ is the average value of the two source-detector distances ρA and ρB. Fig. 2 gives simulation results for the measurement sensitivity distribution in a homogeneous medium, and not the layered structure. I am missing a clear definition of the measurement sensitivity (Equation (2), which is dimensionless?). The scale given in Fig. 2 and Fig. 3 (positive values only) is in mm. For the mapping results, I would like to see results for the actual detector pair separations as experimentally realized.
Response: We have corrected ρA and ρB to ρ1 and ρ2, and added the missing definition of measurement sensitivity. The ΔS of Equation (2) is dimensionless, and the unit of measurement sensitivity is mm.
In Figs. 5 and 6, we have changed to the results for the actual detector distance (ρ2 =5, 8, 14, 30mm) as experimentally realized. The order of (a)-(d) in Figs. 5 and 6 is matched with Figs. 1 and 2.
<Revised manuscript>
(Line 102) In general continuous-wave NIRS, the change in absorbance is Δμa L, and L is the measurement sensitivity[30]. Similarly, the measurement sensitivity in SRS was defined as {LB(x, y, z) – LA(x, y, z)} (mm).
|
Original |
Revised |
|
Fig. 5
|
Fig. 5
|
|
Fig. 6 |
Fig. 6 |
Comment 10: Another aspect is certainly also LED stability which should be discussed for the final application to tissue oximetry.
Response: We have added a description of LED stability to Materials and Methods.
<Revised manuscript>
(Line 142) The oximeter was powered by two AA batteries, and the LED light intensity was stable for 12 hours using a low noise voltage linear regulator and a 30-mA constant current regulator. It was also confirmed that there was no temperature drift of light intensity.
Thank you once again for your consideration of our paper.

Reviewer 2 Report
The subject of the manuscript submitted is of special importance for medical applications. Tissue oximetry has been around for many decades, but also novel developments especially also for pulse oximetry must be mentioned, which tackles in particular the oxygenation of arterial blood. For this also modelling of the radiative transfer in scattering media has been presented, for which also optimal source – detector spacings were determined (see Mehrabi M, Setayeshi S, Ardehali SH, Arabalibeik H (2017) Modeling of diffuse reflectance of light in heterogeneous biological tissue to analysis of the effects of multiple scattering on reflectance pulse oximetry. J Biomed Optics 22 (1): 015004; M. Mehrabi, S. Setayeshi, M. G. Maragheh, S.H. Ardehali, H. Arabalibeik, Design of a new reflectance pulse oximeter by obtaining the optimal source-detector space, Optik 168 (2018) 34–45).
The abstract also should contain all relevant information, for example the wavelengths of the LEDs used for tissue oximetry. As endoscopic applications had been envisaged, much more focus must be given the smaller source-detector configurations, which is not so limiting for skin-subcutaneous tissue applications. This must be more discussed within the introduction. Another important point is that optical parameters of tissue are wavelength dependent, which has not been given with Table 1. In addition, literature must be updated in view of more recent publications with regard to optical constants of skin and layered sections; see the following papers:
Bashkatov AN, Genina EA, Kochubey VI, Tuchin VV. 2005. Optical properties of human skin, subcutaneous and mucous tissues in the wavelength range from 400 to 2000 nm. J Phys D: Appl Phys 38:2543-2555.
T.L. Troy, S.N. Thennadiel, Optical properties of human skin in the near infrared wavelength range of 1000 to 2200 nm, J. Biomed. Opt. 6, 2001, pp.167–176.
Salomatina, B. Jiang, J. Novak, A.N. Yaroslavsky, Optical properties of normal and cancerous human skin in the visible and near-infrared spectral range, J. Biomed. Optics 11, 2006, 064026.
Roggan, J. Beuthan, S. Schründer, G. Müller, ,Diagnostik und Therapie mit dem Laser, Physikalische Blätter 55, 1999, pp. 25–30.
Modeling of radiative transfer with aspects of penetration depths has also been presented many times, and special attention should also be given to
Nasouri B, Murphy TE, Berberoglu H. 2014. Simulation of laser propagation through a three-layer human skin model in the spectral range from 1000 to 1900 nm. J Biomed Opt 19:075003.
Zhang H, Salo D, Kim DM, Komarov S, Tai Y-C, Berezin MY. 2016. Penetration depth of photons in biological tissues from hyperspectral imaging in shortwave infrared in transmission and reflection geometries. J Biomed Opt 21:126006
L.F.A. Douven, G.W. Lucassen, Retrieval of optical properties of skin from measurement and modelling the diffuse reflectance, in: Laser-Tissue Interaction XI: Photochemical, Photothermal, and Photomechanical, Donald D. Duncan, Jeffrey 0. Hollinger, Steven L. Jacques, Editors, Proceedings of SPIE Vol. 3914, 312 -- 323 (2000)
I am missing also the simulation with inhomogenously distributed chromophores as presented in the past (see W. Verkruysse et al., Modelling light distributions of homogeneous versus discrete absorbers in light irradiated turbid media, Phys. Med. Biol. 42 (1997) 51–65; F. Niedorf, H. Jungmann, M. Kietzmann, Noninvasive reflection spectra provide quantitative information about the spatial distribution of skin chromophores, Med. Phys. 32 (5), 1297- 1307 (2005)); due to the vessel structures, the homogenously distributed hemoglobin variants are an approximation which needs testing and which has also been used for the phantom experiments as severe simplification. The authors used a model with a fat-like scattering medium above and below the blood layer, which is not a realistic scenario for skin simulation (see also modeling as presented by Douven and Lucassen above). I am also questioning the two scenarios as given in Fig. 6, which needs discussion and anatomy support.
I think that also the clarity of presentation must be significantly improved and definitions of the terms used be given when used first. For example, on page 3 parameters ρ1 and ρ2 have been presented but not explained, and on page 4 ρ is the average value of the two source-detector distances ρA and ρB. Fig. 2 gives simulation results for the measurement sensitivity distribution in a homogeneous medium, and not the layered structure. I am missing a clear definition of the measurement sensitivity (Equation (2), which is dimensionless?). The scale given in Fig. 2 and Fig. 3 (positive values only) is in mm. For the mapping results, I would like to see results for the actual detector pair separations as experimentally realized.
Another aspect is certainly also LED stability which should be discussed for the final application to tissue oximetry.

Author Response
(Please see the attachment)
RESPONSE TO REVIEWER 2:
We wish to express our appreciation to the reviewers for insightful comments on our paper. The comments have helped us significantly improve the paper. Green texts are the reviewer's comment, and blue letters are the sentences added to the revised manuscript.
Comment 1: There are not clearly summarized conclusions and clinical usefulness of this results.
How optimum source-detector distance or a difference in measurement depth between the skin-subcutaneous and gastric tissues may improve clinical treatment and the usefulness of methods?
How does the measurement of depth variable change the laparoscopy methods and further clinical outcomes of patients?
Response: Thank you for providing these insights. We have added the following clinical usefulness to the discussion; usefulness in skin-subcutaneous tissue measurement, usefulness in gastric tube measurement, improvement by difference in measurement depth between the skin-subcutaneous and gastric tissues.
<Revised manuscript>
(Introduction, Line 55) For example, tissue oximeter has been applied in surgery for gastric tubes [13] and flaps [14], and sensors with source-detector distances of 6 mm and 8 mm were used.
(Discussion, Line 269) The skin-subcutaneous measurement can be applied to monitor the oxygenation of the planer tissue and confirm the therapeutic effect in endovascular therapy for patients with chronic limb-threatening ischemia. The evaluation of blood perfusion can be more reliable by integrating information at several depths. The gastric tube measurement can be used when selecting the gastric anastomosis in surgery for the esophageal cancer. Observation of oxygenation at a depth of 2 mm to 5 mm would lead to a reduction in suture failure due to an accurate assessment of tissue viability. The relatively small difference in measurement depth between the skin-subcutaneous and gastric tissues means that the same sensor probe can be used with a slight depth correction in both applications. This leads to improved convenience and cost reduction by simplifying the device in applications to various tissues.
[13] Fujita, T. Method for evaluating gastrointestinal blood flow and reducing suture failure in esophageal cancer: Blood flow evaluation using regional SO2 (%) and Total Hemoglobin index (Japanese). In Proceedings of the Proc. of the 81st Annual Congress of Japan Surgical Association; 2019; pp. VW02-2.
[14] Tsuge, I.; Enoshiri, T.; Saito, S.; Suzuki, S. A Quick Evaluation of TRAM Flap Viability using Fingerstall-Type Tissue Oximetry. Plast. Reconstr. Surg. Glob. Open 2017, 5, doi:10.1097/GOX.0000000000001494.
Comment 2: The conclusions part is missing….
Response: We have added a conclusion part.
<Revised manuscript>
(Line 280) 5. Conclusion
In this study, the measurement depth of oximetry by spatially resolved near-infrared spectroscopy were examined using Monte Carlo analysis and phantom experiment. From the results of sensitivities for each depth, it was quantitatively found that the measurement sensitivity curve had a peak and the measurement depth can be adjusted by combining the two distances between the light source and the detector. These results suggest the possibility of measuring tissue oxygen saturation at depths of 2-5 mm with selectable measurement depths. The evaluation of tissue viability could be more reliable by observing oxygenation at some depths. Furthermore, the gastric tissue was 10%–20% smaller in measurement depth than the skin-subcutaneous tissue. This means that the same optical probe can be widely applied by correcting the measurement depth according to the target tissue.
Thank you once again for your consideration of our paper.

Round 2
Reviewer 2 Report
The revised manuscript is fine apart from a few minor points, concerning mainly the citation of references (amongst others, use of capital letters within the publication titles should be corrected).
Ref. 3 Author name missing: Takuo Aoyagi, Pulse oximetry: its invention, theory, and future, J Anesth (2003) 17:259—266; DOI 10.1007/s00540-003-0192-6
Ref. 4 format change needed journal name “ Science” only
Ref. 25 Journal abbreviation to be corrected
Refs. 33 and 34 Author names missing
Parameter description needs improvement: Page 3: two parameters were introduced ρ1 and ρ2, which have not been explained, only later on page 4 it is reported to have source-detector distances here, which must be clarified earlier.
Conclusion: In this study, the measurement depths using oximetry by spatially resolved near-infrared spectroscopy were examined …. (grammar - plural)
Author Response
(Please see the attachment.)
RESPONSE TO REVIEWER 2:
I appreciate your confirmation. We have revised the manuscript as follows. Green: reviewer's comment, blue: the added sentences.
Comment 1: The revised manuscript is fine apart from a few minor points, concerning mainly the citation of references (amongst others, use of capital letters within the publication titles should be corrected).
Response: We have corrected the capital letters within the titles of Ref. 6, Ref. 9, Ref. 12, Ref. 13, Ref. 14, Ref. 30.
Comment 2: Ref. 3 Author name missing: Takuo Aoyagi, Pulse oximetry: its invention, theory, and future, J Anesth (2003) 17:259—266; DOI 10.1007/s00540-003-0192-6
Ref. 4 format change needed journal name “ Science” only
Ref. 25 Journal abbreviation to be corrected
Refs. 33 and 34 Author names missing
Response: We have corrected the author names and the journal names as follows. In addition, the information in Ref. 28 was incorrect and has been corrected.
[3] Aoyagi, T. Pulse oximetry: its invention, theory, and future. J. Anesth. 2003, 17, 259–266, doi:10.1007/S00540-003-0192-6.
[4] Shank, C. V.; Ippen, E.P.; Bersohn, R. Time-resolved spectroscopy of hemoglobin and its complexes with subpicosecond optical pulses. Science 1976, 193, 50–51, doi:10.1126/science.935853.
[33] Verkruysse, W.; Lucassen, G.W.; de Boer, J.F.; Smithies, D.J.; Nelson, J.S.; van Gemert, M.J. Modelling light distributions of homogeneous versus discrete absorbers in light irradiated turbid media. Phys. Med. Biol. 1997, 42, 51–65, doi:10.1088/0031-9155/42/1/003.
[34] Niedorf, F.; Jungmann, H.; Kietzmann, M. Noninvasive reflection spectra provide quantitative information about the spatial distribution of skin chromophores. Med. Phys. 2005, 32, 1297–1307, doi:10.1118/1.1851917.
[28] Zaccanti, G.; Taddeucci, A.; Barilli, M.; Bruscaglioni, P.; Martelli, F. Optical properties of biological tissues. Proc. SPIE 1995, 2359, 513–521, doi:10.1117/12.210000.
Comment 3: Parameter description needs improvement: Page 3: two parameters were introduced ρ1 and ρ2, which have not been explained, only later on page 4 it is reported to have source-detector distances here, which must be clarified earlier.
Response: We have added the description of ρ1 and ρ2 after the description of positions A and B.
(Line 101) Here, subscript A corresponds to the light detected near the light source among the two detectors for using the SRS method, and B corresponds to the far side. The distances between the light source and the detectors located at A and B were ρ1 and ρ2, respectively.
Comment 4: Conclusion: In this study, the measurement depths using oximetry by spatially resolved near-infrared spectroscopy were examined …. (grammar - plural)
Response: We corrected the mistake in Conclusion.
(Line 285) In this study, the measurement depth of oximetry by spatially resolved near-infrared spectroscopy was examined using Monte Carlo analysis and phantom experiment.
Thank you once again for your consideration of our paper.
